# Experimental Studies of Microwave Tubes with Components of Electron–Optical and Electrodynamic Systems Implemented Using Novel 3D Additive Technology

**Mikhail D. Proyavin** [1,*] , **Mikhail V. Morozkin** [1] , **Naum S. Ginzburg** [1,2], **Andrej N. Denisenko** [1], **Maxim V. Kamenskiy** [1], **Valentina E. Kotomina** [1], **Vladimir N. Manuilov** [1,2], **Alexey A. Orlovskiy** [1], **Ivan V. Osharin** [1], **Nikolay Y. Peskov** [1,2], **Andrei V. Savilov** [1,2] and **Vladislav Y. Zaslavsky** [1,2]

1   Institute of Applied Physics, Russian Academy of Sciences, 603950 Nizhny Novgorod, Russia
2   Faculty of Radiophysics, Lobachevsky State University, 603950 Nizhny Novgorod, Russia
*   Correspondence: pmd@ipfran.ru

**Abstract:** Novel additive technology of the Chemical Metallization of Photopolymer-based Structures (CMPS) is under active elaboration currently at the IAP RAS (Nizhny Novgorod). The use of this technology has made it possible to implement components of electron–optical and electrodynamic systems for high-power microwave vacuum tubes, such as a gyrotron and a relativistic Cherenkov maser, the design and experimental studies of which are described in this paper. Within the framework of the gyrotron developments, we carried out a simulation of the distribution of the heat load on the collector of high-power technological gyrotron taking into account secondary emission. The prospect of a significant reduction in the maximum power density of the deposited electron beam was shown. The experimental study of the gyrotron collector module manufactured using CMPS technology demonstrated high potential for its further implementation. Recent results of theoretical and experimental studies of a spatially extended Ka-band Cherenkov maser are presented. In this oscillator, the 2D-periodical slow-wave structure made by the proposed technology was applied and a narrow-band generation regime was observed with a sub-GW power level. The design and simulations of a novel selective electrodynamic system for a high-harmonic gyrotron with the planned application of the CMPS technology are discussed.

**Keywords:** gyrotron; electron beams; secondary electron emission; additive technology; surface waves; Cherenkov masers; distributed feedback devices; oscillators; periodic structures; chemical metallization

## 1. Introduction

Recently, significant progress has been made in the development and implementation of high-power sources of electromagnetic radiation in the millimeter and submillimeter wavelength ranges. The most prominent among long-pulse (tens/hundreds of microseconds or more), quasi-continuous and continuous sources are gyrotrons, in which the megawatt power level is reached [1–3]. Among short-pulse (up to tens of nanoseconds) sources, the maximum gigawatt power level was achieved in the sources based on the Cherenkov radiation mechanism (see, for example, [4]). Modern electron complexes based on such sources are becoming an indispensable tool in various fields of science and technology. Microwave complexes based on megawatt power gyrotrons are successfully used for plasma heating in controlled thermonuclear fusion installations [3,5,6] and technological gyrotrons of medium (tens kilowatts) power. For ion sources [7,8] and the high-temperature processing of materials [9,10], gyrotrons operating with relatively low powers (hundreds watt up to kilowatt) at subterahertz frequencies are used in the field of spectroscopy and diagnostics of various environments [11,12]. Powerful pulsed sources are in demand

for topical applications related to the high-gradient particles acceleration at subterahertz frequencies [13,14], the elaboration of pumping systems for compact X-FELs, etc. [15,16].

Progress in the development of most of the mentioned applications is associated with a further power enhancement of the generators and their advancement into the high-frequency regions, which is directly related to the manufacturing technology of various components of these devices. In particular, in the case of high-power gyrotrons for controlled thermonuclear fusion, there is a serious problem of the deposition of a megawatt electron beam on the collector [17]. Currently, a system of longitudinal scanning of the electron beam is used to increase the beam deposition area and a single-stage recovery of residual energy is used to reduce the average thermal load and increase the efficiency of the entire complex [18]. High-frequency beam scanning systems, as well as the multistage recovery of residual electron energy [19,20] are considered for the future. However, these tasks are associated with serious technological difficulties and a lack of experimental data on the dynamics of electrons in the collector region, including the correctness of the secondary emission models used.

The problem of providing a narrow-band generation regime in high-power short-wavelength relativistic masers is associated with the development of high-selective electrodynamic systems. In this regard, small-scale elements in the cavity of low-power sub-THz/THz gyrotrons can improve the electrodynamic selection of the operating type of oscillations [21]. The use of 2D-periodical slow-wave systems realizing a two-dimensional distributed feedback mechanism in Cherenkov oscillators based on intensive relativistic electron beams (REBs) makes it possible to significantly improve the stability of the narrow-band generation regime at the designed mode [22].

Most of the recently implemented microwave devices, in particular gyrotrons, are based on rather "robust" physical principles, which have been allowed to develop over many decades without much attention being paid to the manufacturing technologies. However, advancements in higher frequencies and ultrahigh powers has led to a significantly larger contribution of new physical effects that were previously insignificant and, therefore, have not been investigated in detail. For their study and use in order to improve the operation of the devices, deeper modeling is required, and the refined mathematical models must be verified by comparing the results of calculations with data obtained in experiments. In the case of gyrotrons, improving the properties of the electron beam, namely, minimizing the spread of the transverse electron velocities and increasing the pitch factor (the fraction of the transverse energy of the electrons with respect to the longitudinal one) can significantly increase the efficiency of the electron–wave interaction. In the case of high-power gyrotrons, an important problem arises when calculating the thermal load on the collector under the effect of an intense electron flow. For the most correct calculation, it is necessary to consider the effect of the appearance of secondary electrons in the process of the deposition of the main electron flow on the electrode. When working with high electron energies, the influence of errors in the manufacture of electrodes, the emitting surface, the influence of space charge forces, and so on, increases significantly. If the parameters of the output radiation do not correspond to the calculated ones, it is difficult to understand whether the problem lies in the quality of the electron beam or in the incorrect electron–wave interaction. At the same time, the creation of research microwave complexes requires large amounts of time and financial expenses.

Following the discussed problems, there is a need to create new technologies that make it possible to quickly and inexpensively create experimental samples of microwave devices capable of operating at a high-power level and at high frequencies. Additive technologies are currently spreading into various fields of science and technology, including microwave electronics [23–29]. Various methods are used to metallize 3D-printed mandrels, including magnetron sputtering [30], galvanic methods [31], electroless metal plating with preliminary chemical activation of the surface [32,33], as well as a combination of these methods [34]. The obvious advantages of 3D printing over traditional technologies are its availability, high speed of production, and moderate cost. Another important advantage is

the possibility of one-stage manufacturing of products with a complex shape, including those with internal cavities, in the form of one solid part. While the standard process of manufacturing parts requires the development of design documentation and specialized staff, in the case of additive manufacturing, a 3D computational model can be made using a 3D printer with minimal modifications by the researcher.

At present, the IAP RAS is actively implementing a novel additive technology based on photopolymer 3D printing followed by the chemical deposition of the copper surface layer—the CMPS technology [35,36]. The use of the proposed technology allows the manufacturing of components of a complex shape, including those with small-scale elements, that have a thin (several microns) conductive copper layer on the operating surface. This technology has been examined in a large number of our previous experimental studies at a low power level, in particular, to create prototypes of various components of microwave devices and test their operation in "cold" measurements on network analyzers [36]. However, when operating with an intense electron beam and a high level of radiation power associated with a considerable RF-field strength and thermal loads, as well as in high vacuum conditions, this technology needs to be improved. When it comes to working under conditions of high thermal loads, a rather thick copper coating is required and in most cases the absence of a polymer "skeleton". An alternative to the additive manufacturing of all-metal hollow elements are technologies such as SLS and SLM. Such machines cost millions of dollars, have great difficulty in working with copper, and also have significantly higher surface roughness, which can lead to manufacturing errors. In the present paper, we have focused on the elaboration of the CMPS technology as applied to high-power microwave devices of different classes, such as gyrotrons and Cherenkov masers, when designing and experimentally testing their key components based on this technology.

Accordingly, the paper is organized as follows. In Section 2 we present a project of a gyrotron prototype for studying electron beams, developed using additive technologies. This prototype does not have an e-beam/RF–wave interaction space, its beam forming system (cathode node) is made standard for technological gyrotrons, and the beam transport channel and collector are made using the improved CMPS technology. The results of thermal calculations, tests for vacuum resistance, and testing of a collector section with built-in temperature sensors are presented. Section 3 is devoted to theoretical and experimental studies of a spatially extended Ka-band Cherenkov maser of sub-GW power based on a 2D-periodical slow-wave structure made using the proposed technology. Section 4 describes the design and simulations of a novel selective electrodynamic system for a high-harmonic gyrotron with the planned application of the CMPS technology.

## 2. Electron–Optical System for Technological Gyrotron

### 2.1. Secondary Emission Model

Considering the secondary emission when calculating the operating mode of the collector makes it possible to reduce the power density deposited by the electron flux [37]. However, depending on the model chosen, the collector load power profiles can vary greatly. At the moment, we are considering the following simple qualitative model of secondary emission at the gyrotron collector, based on the experience gained in work with secondary emission in the cathode region and taking into account some other secondary emission models [38,39].

To determine the rules by which secondary electrons are added to the model, consider the incidence of one electron on the collector surface. We introduced the following vectors: $\vec{e_i}$—unit vector codirectional with the electron velocity vector $\vec{v}$; $\vec{n}$— unit vector of the internal normal to the collector surface at the point of electron incidence; $\vec{e_r} = \vec{e_i} - 2\vec{n}(\vec{e_i}, \vec{n})$—unit vector codirectional with the speed of an electron reflected according to the geometric-optical law.

For convenience, in a plane passing through the point of incidence of an electron perpendicular to $\vec{e_r}$ we introduced two unit vectors $\vec{a}$ and $\vec{b}$, which form the local right coordinate system ($\vec{a}$, $\vec{b}$, $\vec{e_r}$):

$$\vec{a} = \frac{[\vec{n}, \vec{e_r}]}{|[\vec{n}, \vec{e_r}]|}, \ \vec{b} = [\vec{e_r}, \vec{a}], \tag{1}$$

Let us set $M \cdot N + 1$ unit vectors $\vec{e_{i,j}}$, defining the directions of movement of secondary electrons in the form of a "fan". The directions of the reflected electrons are $M + 1$ vectors (including $\vec{e_r}$), uniformly distributed along the angle relative to $\vec{e_r}$ in one plane, which are then multiplied in three-dimensional space by rotation around $\vec{e_r}$ $N$ times with a uniform distribution along the azimuth angle. Analytically this array of vectors can be described according to:

$$\vec{e'_{i,j}} = \vec{e_r} + K_i \left( \vec{b} \cdot cos(\beta_j) - \vec{a} \cdot sin(\beta_j) \right)$$
$$\vec{e_{i,j}} = \vec{e'_{i,j}} \Big/ \left| \vec{e'_{i,j}} \right|, \ i = 1, 2 \dots M, \ j = 0, 1 \dots N - 1, \tag{2}$$

To set the currents of secondary electrons, we introduced the distribution function of these particles over the angle with respect to $\vec{e_r}$: $f(\varphi_{i,j}) = cos(k \cdot \varphi_{i,j})$, where $\varphi_{i,j} = acos\left(\vec{e_{i,j}}, \vec{e_r}\right)$ and coefficient $k$ sets the shape of the "hand fan" of secondary electrons. In our calculations, we studied the dynamics of the electron beam in the collector region at $1 \leq k \leq 4$. It is also necessary to consider the current reflection coefficient $R$ which depends on the angle of incidence of the particle: $I_r = I_i \cdot R(\theta)$, where $I_i$—incident large particle current and $I_r$—total current of reflected particles. The current of each secondary particle is given as:

$$I_{i,j} = I_N \cdot f\left(k \cdot acos\left(\vec{e_{i,j}}, \vec{e_r}\right)\right), \tag{3}$$

where $I_N$—normalization coefficient calculated from the relation $\sum_{i,j} I_{i,j} = I_r$.

The electrons are not fully reflected elastically. To do this, we introduced the coefficient $K_v$, such that the speed of secondary electrons $v_r$ and the speed of the incident electron $v_i$ are related by the relation $v_r = K_v \cdot v_i$. In the general case, $K_v$ depends on the electron energy.

In order to interrupt the formation of secondary electrons, we assumed that the electron that completely lands on the surface as soon as the power in its current tube becomes less than the specified value $P_{min}$.

The heat load calculations of the collector were made using our own three-dimensional program code based on the IBSimu library [40]. Based on the results of calculations of the system considering secondary electrons, the following conclusions were made regarding the selection of model parameters: firstly, the minimum power $P_{min}$ required for the formation of secondary electrons should not be too small, since this leads to the smearing of peaks and the appearance of long unnecessary "tails", and also greatly reduces the counting rate. Secondly, there is no need to specify many secondary electrons; one in the direction $\vec{e_r}$ plus 3–5 electrons at an angle to it is enough. Thirdly, the value of $K_v$ has a more significant effect on the trace of the electron beam, although much weaker than just considering the secondary emission.

Based on such a selection of model parameters, qualitative results were obtained on the change in power density due to the consideration of the secondary emission of electrons from the collector surface of the technological gyrotron with an electron beam current I = 2.4 A and an accelerating voltage U = 23 kV. When secondary electrons are considered, the trace of the electron beam lengthens several times. This requires a significant lengthening of both the entire collector and its cooling zone. At the same time, the peak thermal load on the collector is significantly reduced, which opens up the possibility of using smaller

diameter collectors in gyro devices. The calculated power density profile on the collector is shown in Figure 1. The beam trace length was about 5 cm without secondary emission and increased to about 15 cm when secondary particles were considered.

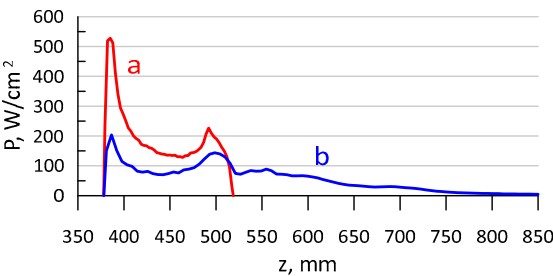

**Figure 1.** Power density dissipated in the collector: (**a**) without secondary emission and (**b**) with secondary emission taken into account.

## 2.2. Designing and Testing the Electron–Optical System for the Technological Gyrotron

The technological 28 GHz gyrotron driven by 2.4 A/23 kV electron beam was taken as the basis for the future complex. As a part of this task, the nodes for connecting parts of the gyrotron were developed, considering the restrictions on installation in a magnetic system and the organization of cooling channels. First, the connection point of the cathode leg and the details of the channel for transporting the electron beam to the gyrotron cavity were worked out. To avoid the effect of generation, the cavity was replaced by a structure of complex geometry, which suppressed the excitation of the RF-field in the beam transport channel. We used the COMSOL Multiphysics program [41] to calculate the temperature distribution in the proposed configuration of the connecting node to take into account thermal radiation from the cathode. We also calculated the mechanical stresses created when the polymer is joined to the metal. The simulation results are shown in Figure 2.

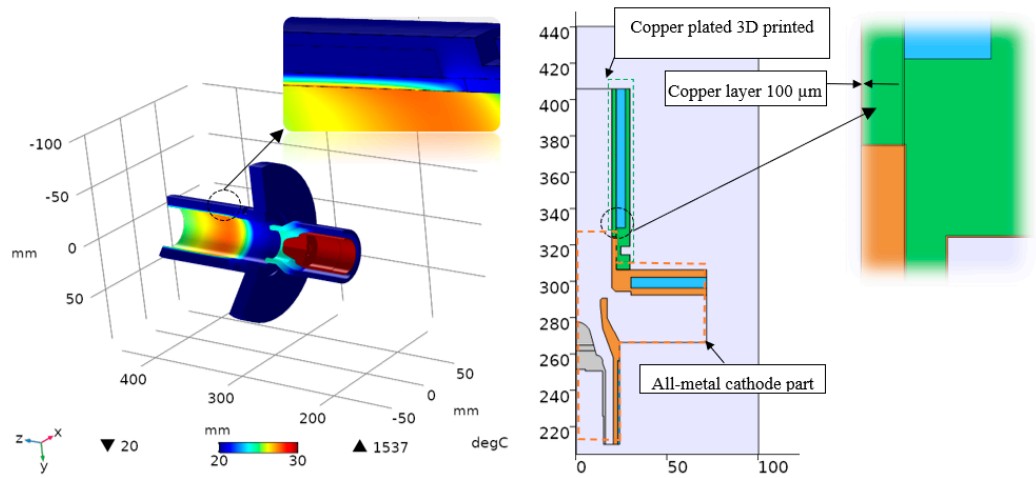

**Figure 2.** Temperature distribution in the cathode-anode unit: green indicates a 3D printed photopolymer part, orange indicates a copper solid part or a copper coating of a polymer part, and blue indicates a water-cooling channel.

According to the simulations, the main thermal heating was due to the thermal radiation of the cathode itself, the temperature of which was about 1500 °C. Obviously, the copper-clad photopolymer part was the most vulnerable from the point of heating due to the poor thermal conductivity of the plastic, which was observed during the modeling process. In Figure 3 a graph of the temperature on the surface of the most heated areas is shown. The maximum temperature did not exceed 28 °C, which was achieved due to the embedded cooling system.

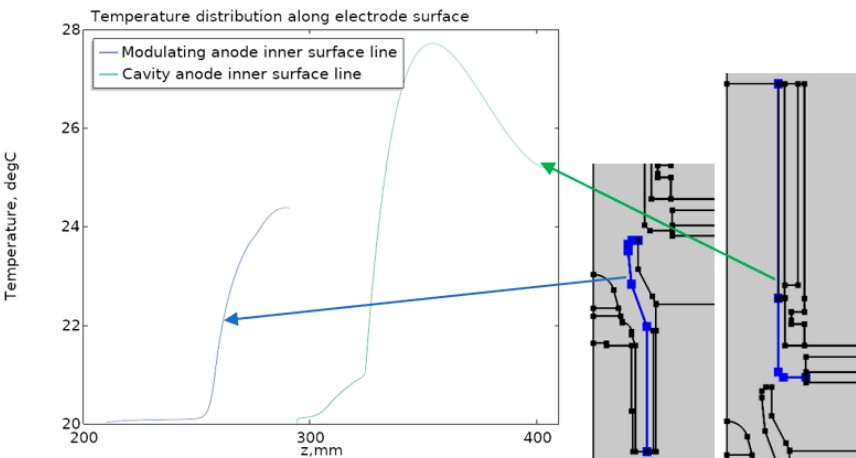

**Figure 3.** Temperature profile on the inner surface of the hottest elements.

In its absence, the maximum temperature of the parts was significantly higher. Despite the low thermal load in the printed part, the thickness of the photopolymer in the region of maximum temperature was minimized due to the low thermal conductivity of the photopolymer. The thickness of the copper coating was chosen based on the results of an experiment to study the vacuum resistance of a copper-plated printed model (see Figure 4) with a large (about 1 L) internal volume. In the course of this experiment, a HiCube 80 pumping station with a capacity of 67 L/s was used and the test model was connected to the vacuum system in the configuration proposed for connecting the cathode part to the HEB transport element. The reached limit of the vacuum level of the pump itself ($10^{-7}$ Torr) in the test model with a copper coating thickness of about 200 μm demonstrated the successful testing of both the connection itself and the copper plating technology. Note that in this case, after chemical metallization, a denser layer of copper was galvanically grown.

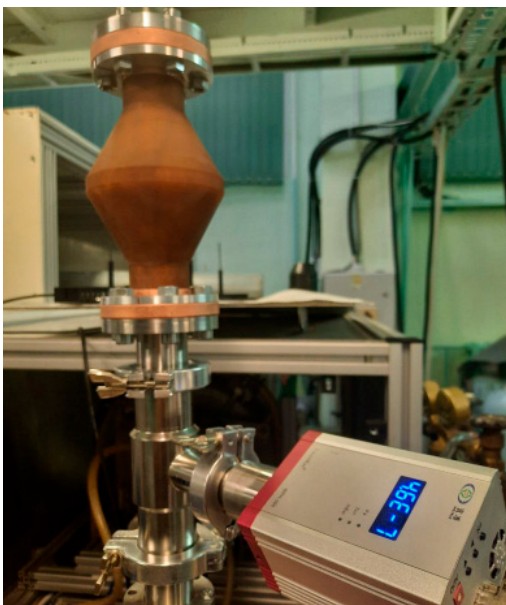

**Figure 4.** Vacuum testing of a test piece with an internal copper coating thickness of about 200 μm and an internal volume of about 1 L.

Considering the limitation of the dimensions of parts created on the existing 3D printer (400 mm), as well as the need to install the gyrotron in a magnet with a bore of 60 mm, the collector area was divided into three parts (see Figure 5). The first part is a transition from

the cavity region, which has a smaller diameter of the connection node. On the other side, the connection diameter of the transition element is larger and corresponds to the increased transverse dimension of the collector. Due to absence of a power load, this element is covered by thin copper layer only to provide good vacuum conditions (about 100 μm).

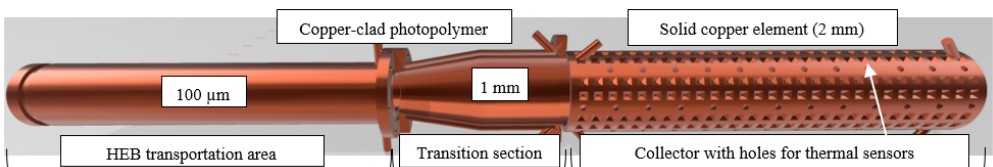

**Figure 5.** Scheme of the 3D printed part of the research gyrotron.

Based on the design conditions of current deposition on the collector and the desire to operate the 3D-printed collector in continuous mode with a HEB power of up to 55 kW, the diameter was chosen to be 66 mm. To assess the capabilities of the cooling system, the calculation of the thermal load on the collector was carried out with the calculated trace of the HEB on the collector in a single pulse mode (pulse length 1 s) for two options: copper-plated (layer thickness 1 mm) and polymer and all-metal (2 mm). The calculation results are shown in Figure 6.

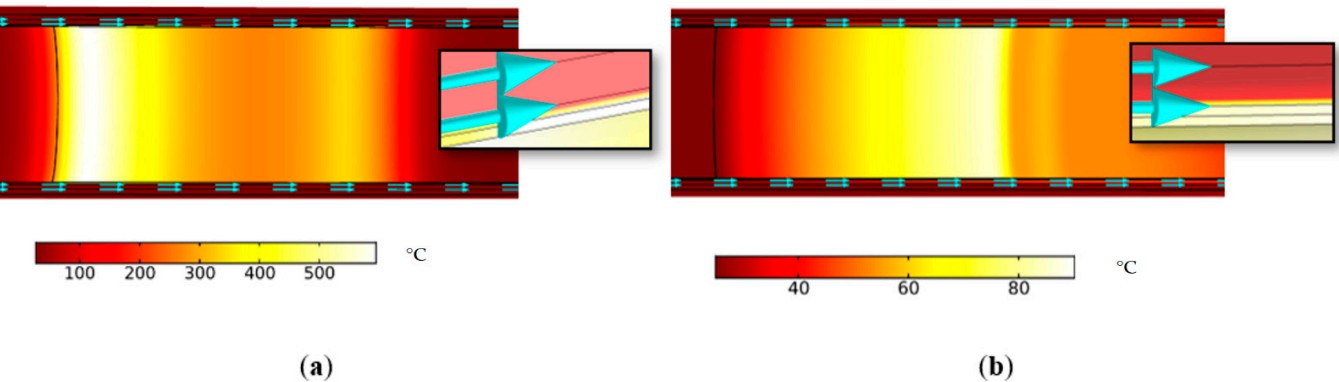

**Figure 6.** Thermal trace of the beam (**a**) in the presence of a polymer layer between the copper collector and the water-cooling system and (**b**) without the polymer layer.

In the case of a polymer layer between the inner copper wall and the cooling channel, its temperature rose to 600 °C at the end of the pulse (Figure 7, top), while on the outer side of the photopolymer it was only 32 °C. Such a temperature gradient meant extremely inefficient heat transfer and melting of the photopolymer during operation. In the case of an all-copper collector, the temperatures of the copper wall inside and outside were almost equal and were within 80 °C (Figure 7, bottom). This was ensured by the flow of water with an initial temperature of 25 °C, which, during the pulse, increased to 28 °C with the flow speed of 1 m/s. The heating of the system at the time of 0.5 s and 1 s (the end of the pulse) was practically the same, which meant that it was close to the stationary regime. Therefore, the collector structure must be all-metal and have direct contact with the cooling water. This means that the collector production technology should involve the further removal of the photopolymer base, which serves as an obstacle to the outflow of heat from the copper layer heated by the beam to the coolant. To manufacture a collector in this way, it is necessary to modify the CMPS technology to make it possible to remove the photopolymer preform from the internal volume, which has a complex surface shape.

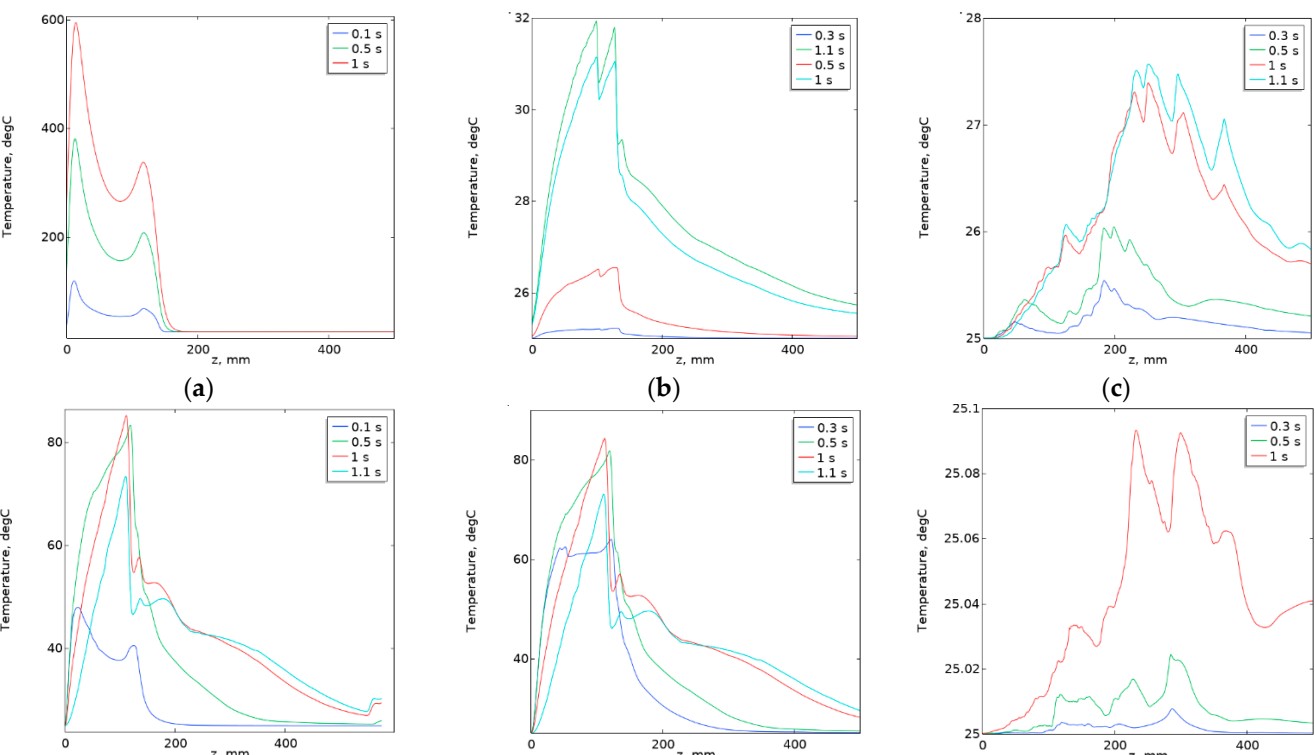

**Figure 7.** Temperature profiles on a metalized copper collector with a polymer layer (**top**) and an all-copper collector (**bottom**): (**a**) on the inner copper surface, (**b**) the outer surface of the copper layer at the boundary with water, and (**c**) the outer surface of the water-cooling channel.

To control the profile of the electron beam trace, the geometry of the photopolymer preform for the collector involved the installation of an array of high-precision digital temperature sensors Smartec SMT172 with TO18 metal case. These sensors were chosen because of their high measurement speed, so at a temperature of 25 °C the measurement time was 1.8 ms with a measurement accuracy of about 0.25 °C. With an increase in the measurement time to 0.1 s, the accuracy of the sensor was about 0.01 °C. The sensors were located on the surface of the collector in a spiral (see Figure 8). This arrangement made it possible to measure the two-dimensional density distribution deposited on the reservoir with a spatial resolution of about 30 mm. After adjusting the lamp in a magnetic field, the power density should become almost axisymmetric, i.e., depending only on the longitudinal coordinate. In this case, the arrangement of the sensors in a spiral made it possible to obtain a higher resolution of the one-dimensional power distribution from the longitudinal coordinate, since the *z* distance between adjacent sensors was only 3–4 mm.

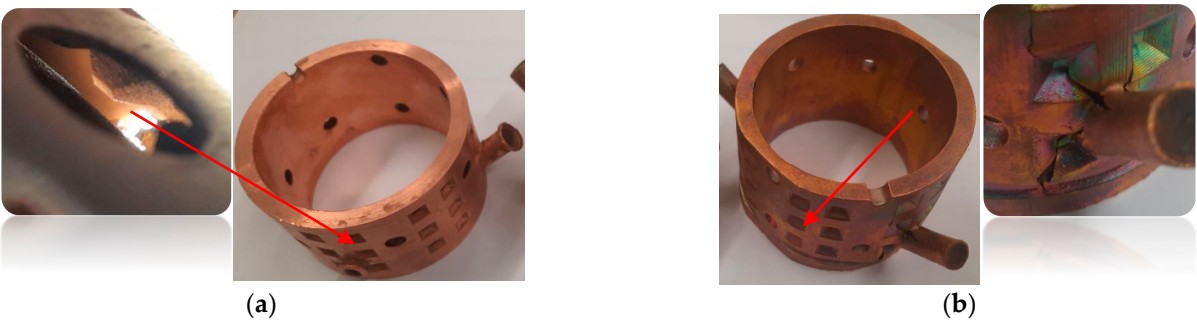

**Figure 8.** Manufactured sectors of the collector: (**a**) considering the difference in TEC and (**b**) without.

The manufacturing technology of the collector with sensors was as follows: after the mandrel (created on Phrozen brand printers with an 8 K resolution matrix) was metallized, the thickness of the copper layer was increased to 1 mm and polymer "skeleton" was deleted. Then, sensors in photopolymer sleeves with a metallized end on the inside of the collector were installed into the holes for temperature sensors, with the help of which the trace of the electron flow was studied. Then, to create a whole dense layer on the inside of the collector, another 1 mm of copper was added.

The experimental development of this technology was divided into two stages. In the first stage, a sector of this collector was printed from a special photopolymer. A feature of this resin is the possibility of its complete evaporation at a temperature of more than 300 °C without the formation of combustion products. At the same time, at a temperature of 270 °C, this photopolymer becomes liquid and can be removed from the metallized body. However, the thermal expansion coefficient (TEC) of this polymer is much higher than that of copper. As a result, during the experiment, when we heated the model to 230 °C, the copper layer was destroyed in several places due to the mechanical stresses caused by the expansion of the photopolymer. To reduce the impact of the difference in thermal expansion coefficients, both the 3D model itself and the process of extracting the photopolymer base were modified. As a result, it was possible to manufacture an all-metal sector of the gyrotron collector with holes for thermal sensors, cooling channels, and a profile for the turbulent movement of the water flow (Figure 9).

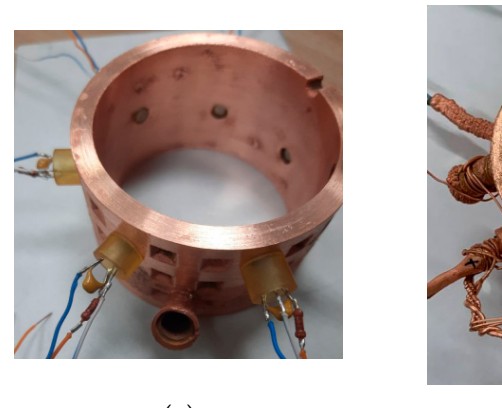 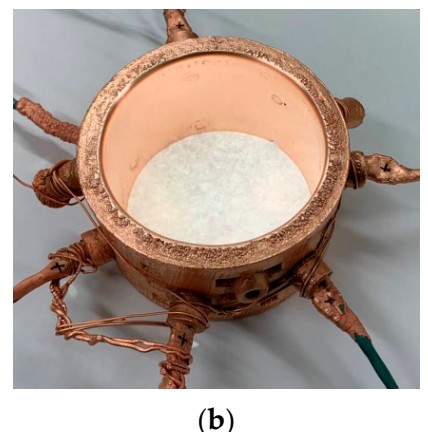 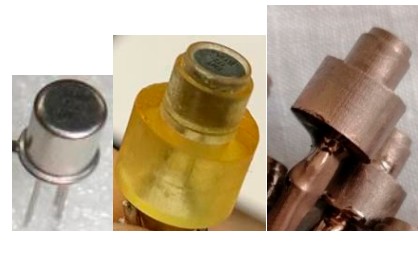

(**a**)  (**b**)  (**c**)

**Figure 9.** Collector sections having copper thickness of (**a**) 1 mm and (**b**) 2 mm; (**c**) installed heat-insulated thermal sensors.

For thermal insulation, the temperature sensors were installed in printed polymer sleeves, which had a body geometry and fitted tightly into the collector body. Then further galvanic build-up was carried out. Thus, it turned out that the conductive end of the sensors through the copper layer received information about the heating of the collector in a certain coordinate. At the same time, heat leakage from other sides of the sensor, bordering on the cooling circuit, was minimal.

This section of the collector was experimentally tested for operability in the process of heating its inner surface with a profiled heat source, which was a nozzle on a hot air gun with a temperature control. During the experiment, two temperature profiles were set, with one and with two maximums. The temperature difference between the initial (room) and stationary temperatures was measured at each sensor during heating. Despite the negative factors, such as: (1) a small number of sensors (7 pcs.); (2) narrow, about 7 mm, axially symmetrical air duct; and (3) high thermal conductivity of copper, blurring the distribution, the resolution of the thermal sensors was sufficient to clearly observe the extremes of the heat source profile. The results are shown in Figure 10.

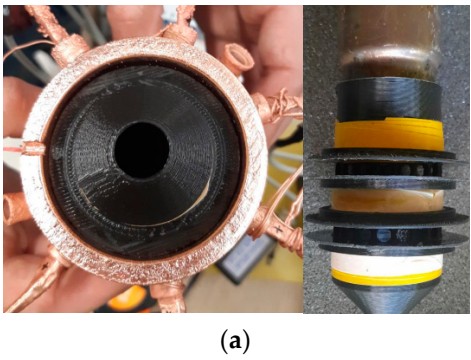
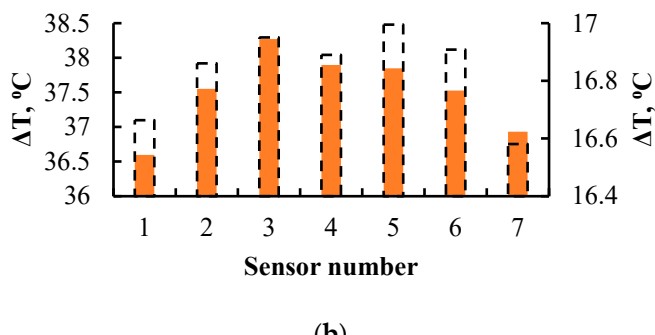

(**a**)                                    (**b**)

**Figure 10.** (**a**) Gyrotron collector sector with a heat source profiled along the longitudinal axis. (**b**) The results of the temperature distribution along the inner wall of the collector sector with a single-hump (solid) and two-hump (dash) profile of the heat source.

## 3. Sub-GW Power Ka-Band Cherenkov Oscillator with a 2D-Periodical Slow-Wave Structure

As noted above, relativistic generators based on high-current REBs currently provide a record level of pulsed power in the centimeter wavelength band [4]. However, a further increase in the output power and their advance into shorter radiation wavelength bands leads inevitably to the necessity of increasing oversized interaction space, and, correspondingly, to the problem of the provision of mode selection with respect to the transverse index. Ensuring coherence of radiation in a strongly oversized generator can be achieved by using a novel feedback mechanism—the so-called two-dimensional distributed feedback (2D DFB) [34]. Previous theoretical investigations have demonstrated a large potential of the 2D DFB mechanism for obtaining powerful narrow-band radiation in various types of relativistic masers fed by REBs with a transverse size which in order of magnitude (or more) should be larger than the wavelength. Operability of the novel feedback mechanism was demonstrated experimentally in W-band free electron maser (FEM) at an interaction space transverse size (width) of up to 50 wavelengths with an output power level of 50–100 MW [42,43].

A promising development of the 2D DFB concept is the elaboration of the planar relativistic oscillators based on the Cherenkov mechanism of the electron–wave interaction. Unlike FEMs, these oscillators involve the radiation from rectilinearly moving electron beams, which conceptually simplifies the formation system and the electron optics system and increases the tolerances on the particle parameters spread. As a result, the Cherenkov devices make it possible to use more intense REBs as compared to FEMs and thereby increase the output radiation power.

In the case of Cherenkov masers, the 2D DFB is implemented in 2D-periodical slow-wave structures [22]. Such structures form slow surface waves responsible for an efficient interaction with a rectilinear REB and, simultaneously, provide, at any size of the system, for a set spatial field structure along the coordinate directed in the normal direction to the corrugation. At the same time, along the second ("wide") transverse coordinate directed along the corrugation, the synchronization of radiation from spatially extended REBs is achieved by means of transverse (with respect to the direction of particles motion) wave fluxes arising in the described structures. As a result of using such approaches, a stable narrow-band oscillation regime is achieved under condition of substantially oversized interaction space.

A spatially extended relativistic maser of the Cherenkov type operating at the Ka-band was constructed based on the high-current explosive-emission accelerator "Sinus-6" 0.5 MeV/5 kA/25 ns (IAP RAS). The so-called π-mode operation regime was chosen for this oscillator (such oscillators are also commonly referred to as surface wave oscillators, SWO), in which a slow fundamental harmonic is exploited for increasing the electron beam

coupling impedance. For this Ka-band SWO, the 2D slow-wave structure of cylindrical geometry was designed with an average diameter Ø = 44.3 mm (perimeter of about 16 wavelengths) and a length of about 16 cm with a 2D sinusoidal corrugation of 7.75 mm period, 3.5 mm depth, and 16 azimuthal turns (Figure 11). In the 2D SWO, the slow-wave structure combines the properties of a slow-wave system realizing the conditions for an effective Cherenkov interaction with a high-current rectilinear REB, and a high-Q resonator utilizing the mechanism of 2D DFB and providing selective excitation of the operating mode in the strongly oversized system. The considered scheme of the oscillator is characterized by the presence of the four electromagnetic (e.m.) energy fluxes propagating in the axial $\pm z$ and the transverse (azimuthal) $\pm \varphi$ directions. To provide a single output of radiation from the 2D SWO, an additional coaxial reflector was designed for installation at the up-stream side (cathode-side) of the generator.

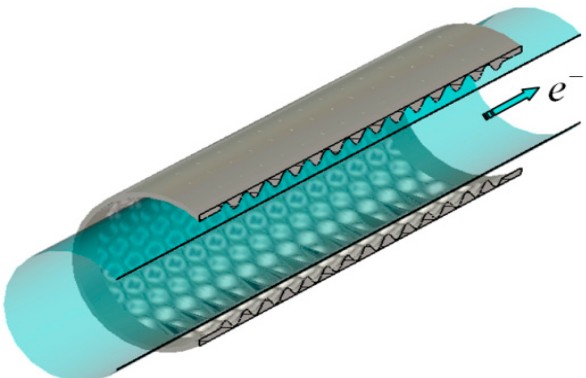

**Figure 11.** Schematic of a Cherenkov SWO with a tubular REB and cylindrical 2D slow-wave structure realizing the 2D DFB mechanism.

Simulations of 2D SWO based on the "Sinus-6" accelerator were carried out using PIC code CST Studio Suite. Parameters for the simulations were taken close to the experimental conditions. The results of simulations are presented in Figure 12 and demonstrate the establishment of a narrow-band oscillation regime under the design parameters. The output power reached 0.5 GW under the electron efficiency of ~25% (Figure 12a). Longitudinal and transverse structures of the various components of the RF-field inside the interaction space of the oscillator are shown in Figure 12b. Components $H_\varphi$ and $E_z$ corresponded to the interference of the axially propagating partial waves interacting with a rectilinear tubular REB, and the $H_z$ field component corresponded to the interference of transversely (azimuthally) propagating partial wave-beams ("waves of synchronization"). Simulations demonstrated that the RF-field structure corresponded to the excitation of the fundamental mode of a 2D-periodic slow-wave structure. At the same time, the spatial distribution of the partial wave synchronous to the electron beam (emitted from the SWO) had an azimuthally symmetric structure and, when decomposed by proper waves of a cylindrical waveguide, contained the waveguide modes of the $TM_{0,n}$ type.

For the implementation of the Ka-band SWO at the "Sinus-6" accelerator facility, electron–optical experiments on the formation of a wide tubular REB with a diameter of about 40 mm were conducted and the stable transportation of this beam through the interaction space was achieved in the solenoid with a guide field of ~1.4 T. In the proof-of-principal experiments, powerful narrow-band radiation was detected under the design parameters. Typical oscilloscope traces of accelerating voltage and RF-pulse are presented in Figure 13a. An output radiation directional diagram was analyzed by means of the neon bulb panel set at various distances from the oscillator output window (see Figure 13b) and demonstrated a pronounced minimum at the axis, which corresponded to the excitation of designed pattern of $TM_{0,n}$-type modes. The output power (according to colorimetric diagnostic) was measured at the level of ~200–250 MW.

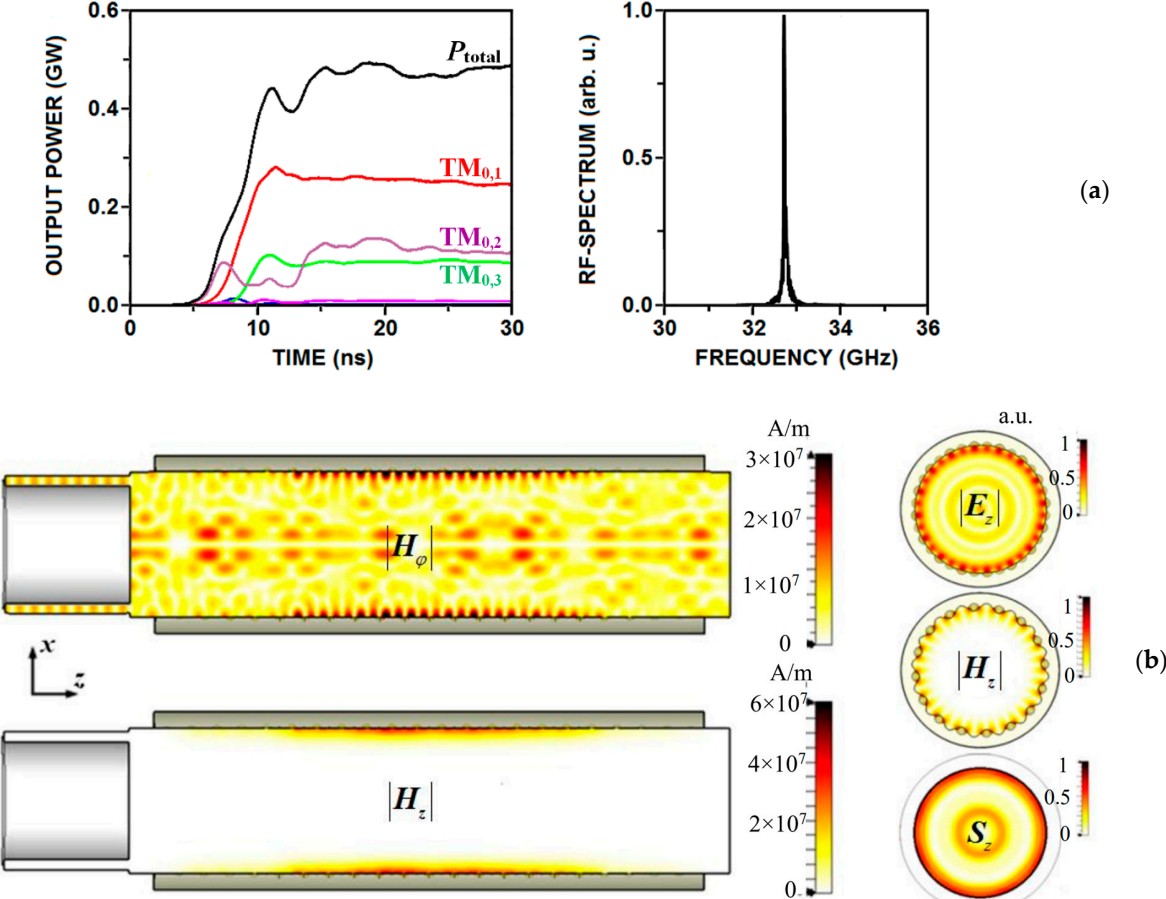

**Figure 12.** Results of the 3D PIC simulations of Ka-band 2D SWO based on the "Sinus-6" accelerator (code CST Studio Suite). (**a**) Time dependence of the total output power and partial powers associated with different waveguide modes (**left**) and radiation spectrum in the steady-state generation regime (**right**) as well as (**b**) spatial instantaneous structure of the RF-field: $H_\varphi$ and $H_z$ components in the longitudinal cross-section (**left**) and $E_z$ and $H_z$ components in the transverse cross-section inside the interaction space and axial component of the Poynting vector $S_z$ of the output radiation (**right**).

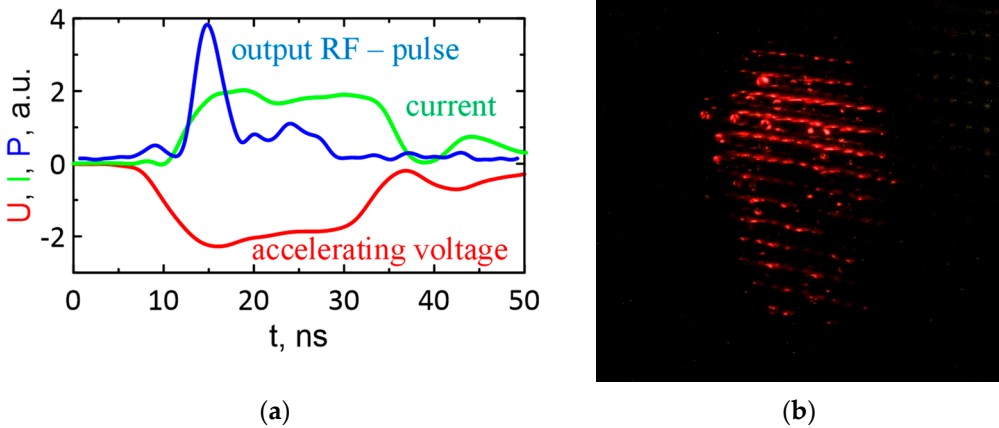

**Figure 13.** Results of the experimental studies of the Ka-band SWO with 2D DFB based on the "Sinus-6" accelerator: (**a**) typical oscilloscope traces of accelerating voltage (red line), electron beam current (green line), and output RF-pulse (blue line) and (**b**) photograph of glowing neon bulb panel under action of a generated RF-pulse (distance to the output window ~1.5 m).

One of the main research issues in the conducted experiments was the comparison of two different technologies for manufacturing prototypes of slow-wave structures. Figure 14 presents photographs of the prototypes manufactured utilizing "conventional" copper electroplating technology (production of a duralumin mandrel on a multi-coordinate milling machine with subsequent electrochemical build-up of copper and further chemical etching of the mandrel) (Figure 14a) and novel additive technology (Figure 14b). In this case, within the framework of the CMPS technology, a thin copper layer (about 10 microns) was first chemically coated on a photopolymer mandrel made on a 3D printer, and then the working layer (about 1–2 mm) was built-up by electroplating, followed by thermal melting of the mandrel. Then, by the method described in Section 2.2 (manufacturing of the part of gyrotron's collector), using the procedure of temperature oscillation with high amplitude, we deleted the polymer "skeleton". The experiments show that when using slow-wave structures with the same geometric parameters of corrugations, identical oscillation regimes and a similar output power level were observed. This confirms the high prospects for the application of the novel CMPS technology, which has a lower cost and provides a higher speed of production.

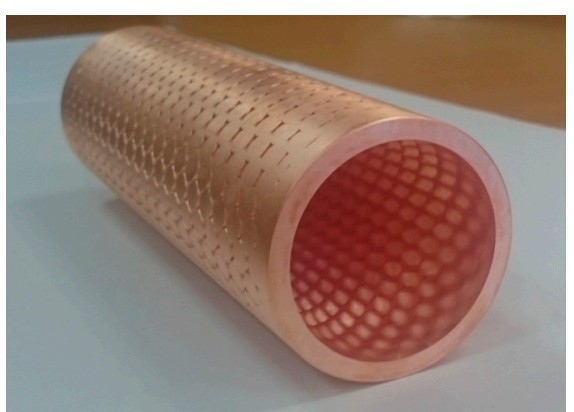

(**a**)

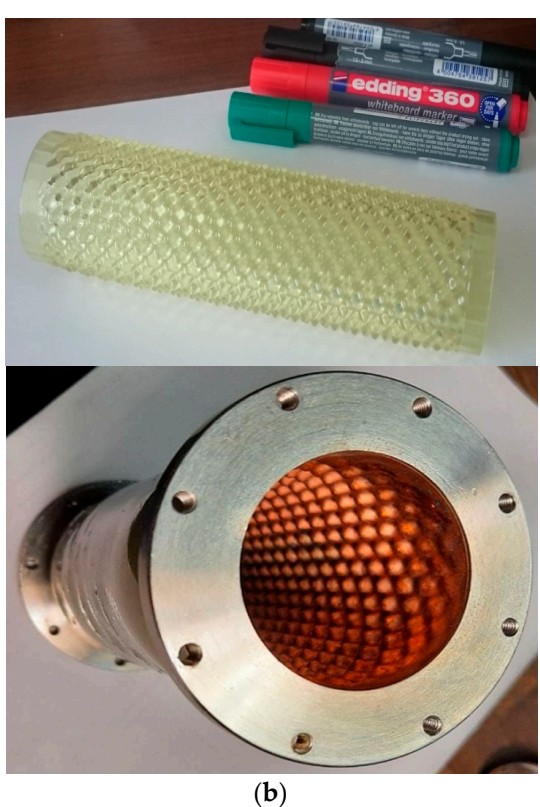

(**b**)

**Figure 14.** Photographs of the prototypes of the 2D-periodical slow-wave structures for Ka-band SWO manufactured using different technologies: (**a**) "conventional" copper electroplating technology and (**b**) novel CMPS technology (**upper**—the photopolymer 3D printed mandrel, **lower**—the 2D slow-wave structure after coating the copper layer and melting the mandrel).

## 4. Selective Resonator for a High-Harmonic Gyrotron

Another example of using novel additive CMPS technology is related to the elaboration of a selective cavity of a new type for gyrotrons operating at high cyclotron harmonics. As a method to significantly increase the selectivity of excitation of operating modes at high cyclotron harmonics in gyrotrons of the millimeter and submillimeter ranges, it was proposed to use a cavity whose cross-section was close to axisymmetric (round), but at the same time had one or more azimuthal inhomogeneities (Figure 15) [21]. It is important that such inhomogeneities have a resonant character only for one operating mode. The

meaning of the term "resonant character" used here is illustrated in Figure 15 a. The azimuthal component of the resonator's own mode was zero on its wall. This meant that if the operating mode of the gyrotron is the near-cutoff mode $TE_{m,n}$, then the radius of the inhomogeneity is chosen so that it corresponds to the near-cutoff mode $TE_{m,n+1}$ at the same frequency. In fact, an increase in the radius in the inhomogeneity corresponds to one additional radial variation of the wave field at a given frequency. As a result, the operating $TE_{m,n}$ mode is transformed inside the inhomogeneity it into the $TE_{m,n+1}$ without losses and reflections. Therefore, the operating mode practically "does not feel" this inhomogeneity, and the excitation of this wave occurs in the same way as in a regular circular cavity. In fact, the cavity with "proper" azimuthal inhomogeneities represents a resonant structure for a single pair of modes $TE_{m,n}$ + $TE_{m,n+1}$.

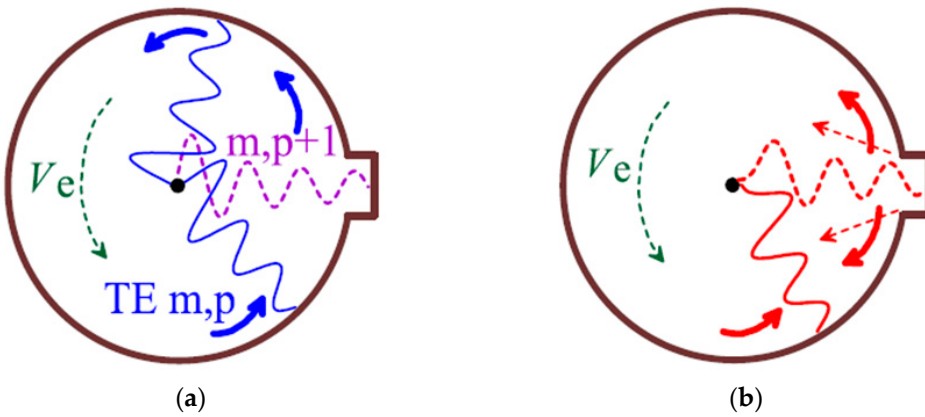

**Figure 15.** Cross-section of a cavity with one azimuthal inhomogeneity: (**a**) rotating operating wave and (**b**) transformation of rotating parasitic wave.

At the same time, for parasitic waves, the presence of this inhomogeneity means that the cavity ceases to have the property of azimuthal symmetry (Figure 15b). As a result, a number of factors (stopping wave rotation and converting a circularly polarized wave into two standing waves, distortion of the wave transverse structure, and an increase in Ohmic losses) leads to a noticeable increase in the starting currents of the parasitic waves [21].

Currently, a cavity has been designed and manufactured based on 3D photopolymer printing technology for a demonstration experiment illustrating the selective abilities of azimuthally asymmetric electrodynamic systems based on a technological gyrotron operating at a frequency of about 28 GHz on the second cyclotron harmonic, described in chapter 2. Earlier, the selective excitation of the third harmonic was obtained in this gyrotron only under conditions of reflection from an external load. Now, to solve the selectivity problem, it was proposed to use an operating cavity with a selective element in the form of an azimuthal groove parallel to the axis of the system. The gyrotron cavity is the most sensitive part of the generator in terms of the manufacturing error. In the standard form, the gyrotron cavity has a simple cylindrical shape, which greatly simplifies the process of its manufacture on a CNC machine. If it is necessary to create a resonator with an azimuthally asymmetric shape, the technology changes significantly. It is necessary to make a negative of a real shape from aluminum with high accuracy, then build up a thick layer of copper, perform surface treatment, and etch the metal blank. All this is significantly inferior to the process of creating such an element using the CMPS method.

According to the simulations, if we used the traditional gyrotron cavity with the circular azimuthally symmetric cross-section, then the tubular electron beam excited the rotating mode $TE_{1,1}$ at the fundamental cyclotron resonance. In a wide range of magnetic fields, it was suppressed with the wave excited at higher cyclotron harmonic (Figure 16). However, if the excitation of the mode $TE_{1,1}$ was suppressed, then we could expect the excitation of other modes, in particular, the mode $TE_{0,2}$ at the third cyclotron harmonic. Actually, according to Figure 16, in the magnetic field range close to 6.9 T, in this case only

the mode $TE_{0,2}$ was excited at operating currents 1–2 A. In the absence of the mode $TE_{1,1}$, she had no other competitors. The numerical simulations below confirm this statement.

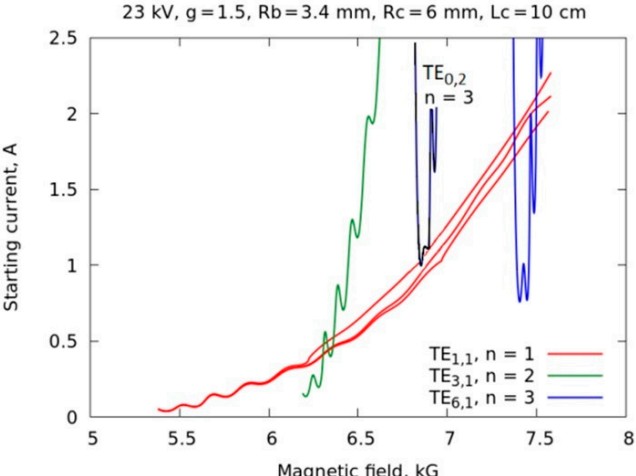

**Figure 16.** Results of the simulations of a technological gyrotron (beam voltage 23 kV, pitch factor $g = 1.5$, beam radius $R_b = 3.4$ mm, cavity radius $R_c = 64$ mm, and length $L_c = 10$ cm). Starting currents of modes on different cyclotron harmonics.

We planned to prove this using a cavity with one azimuthal irregularity (groove). The 3D model of this cavity is shown in Figure 17. The radius of the groove was chosen from the condition that the cutoff frequency of the mode $TE_{0,3}$ corresponding to this radius coincided with the cutoff frequency of the mode $TE_{0,2}$ of the original circular waveguide. The irregularity had the shape of a "boat". This was completed in order to align the ends of the cavity with round conical sections and to exclude the transformation of the operating mode $TE_{0,2}$ into other modes. For this, transition regions were provided in which the angular size of the groove sector decreased linearly to the ends of the cavity.

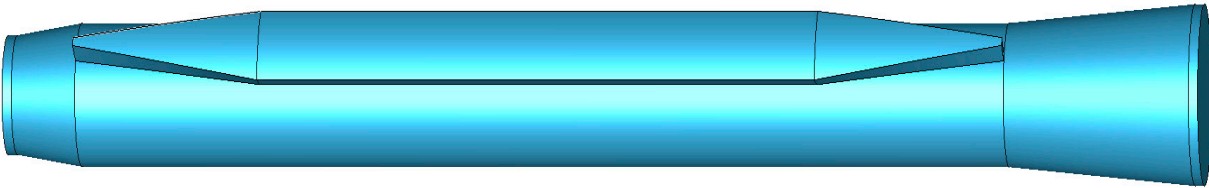

**Figure 17.** Schematic of a gyrotron cavity with an azimuthal groove.

Note that in the scheme described in [21] a cavity with two azimuthal irregularities was proposed. The fact is that, as a rule, an azimuthally asymmetric transverse mode with the "correct" direction of its rotation is used as an operating mode in gyrotrons. In this case, the two grooves (located in places corresponding to the zero and maximum of the field of the standing component of the rotating operating mode) are needed in order to prevent the rotation of the operating wave from stopping. In contrast to the study in [21], in this paper we used only one groove to improve the mode selection. This was a consequence of the advantage of the operating azimuthally symmetric mode $TE_{0,2}$ we chose; there was no problem stopping rotation for it.

The 3D PIC simulations of this system were carried out using the CST Studio package. A Tesla K80 GPU was used to speed up calculations. The electron energy in the input electron beam was 23 kV and the pitch factor was $g = 1.5$ with a spread in electron velocities taken into account (the width of the Gaussian transverse velocity distribution function was 20%, which is typical for this type of gyrotron gun). When using the proposed asymmetric cavity for the $TE_{0,2}$ mode in a wide range of parameters (beam current and magnetic

fields corresponding to the generation zone in the single-mode approximation), the stable generation of radiation at the $TE_{0,2}$ mode on the third cyclotron harmonic at a frequency of about 60 GHz was obtained. Figure 18 shows the results of the numerical simulation for a constant magnetic field of 0.69 T and an electron beam current of 1.4 A.

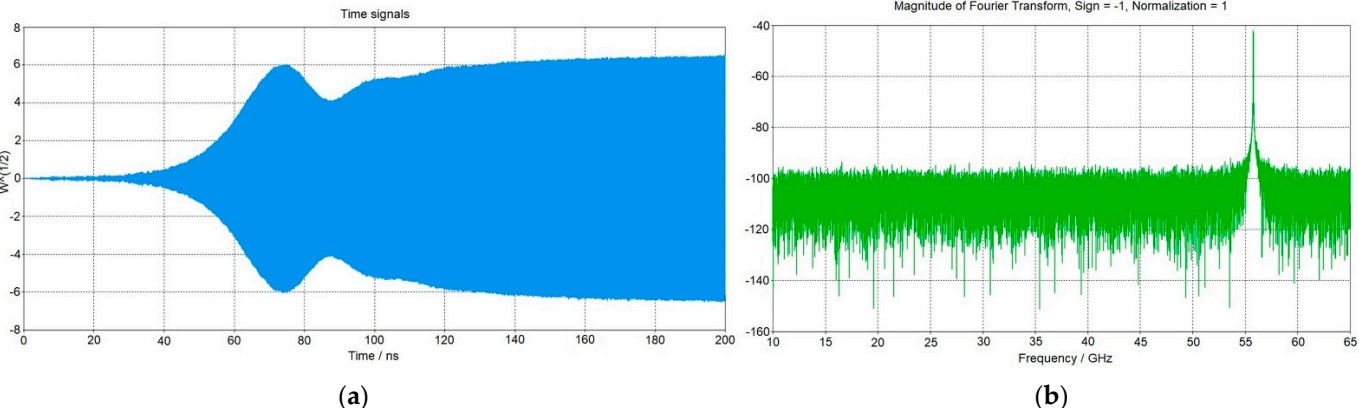

(**a**)                                          (**b**)

**Figure 18.** Results of the 3D CST simulations of a V-band gyrotron with a cavity and with a selective cavity with an azimuthal groove: (**a**) time dependence of the amplitude of the output signal demonstrating the establishment of the stationary regime of generation (the square of the amplitude corresponds to the instantaneous power in watts); (**b**) frequency spectrum of the field inside the cavity—only the third harmonic of the cyclotron frequency was excited.

Currently, the design development of the described resonator has been carried out and manufacturing of its prototype using CMPS technology is in progress. The system is being assembled to conduct experiments on testing this gyrotron.

## 5. Conclusions

The article presented the use of CMPS technology to create microwave components for high-power microwave devices of current interest—gyrotrons and Cherenkov masers. In the first part of the work, a model of a gyrotron was proposed to study the properties of helical electron beams formed by a classical magnetron-injection gun, and in particular, the secondary emission effects. A simple model of secondary electron emission in the collector region of the gyrotron was proposed. Modification of the CMPS technology made it possible to manufacture all-copper elements, including single parts with hidden cavities and channels. Thus, a section of the gyrotron collector with a cooling channel and integrated temperature sensors was made. The heating of such a collector was studied numerically and the high spatial resolution of the array of sensors was experimentally shown. In the last part of the work, a new selective electrodynamic system for a high-frequency gyrotron was proposed, which is currently being implemented using novel CMPS technology.

The possibility of using CMPS technology to create electrodynamic systems of pulse generators operating at a multi-megawatt power level was demonstrated in the second part of the work. A Cherenkov maser based on the Bragg structure with a complex surface shape made using this technology was realized at the Ka-band. Successful experiments in conditions of high-current electron beams, high intensity of the RF-fields, and a relatively strong vacuum confirmed the fine quality of the copper layer of the product. Good correspondence of the theoretical analysis to the experimental results indicated a high accuracy in relation to the manufacturing of the developed microwave components for the specified frequency range.

**Author Contributions:** Conceptualization, M.D.P., N.Y.P. and A.V.S.; methodology, M.D.P., M.V.M., V.E.K., N.Y.P., V.Y.Z., V.N.M. and A.V.S.; software, M.V.M., V.Y.Z. and I.V.O.; validation, M.D.P., N.S.G. and A.V.S.; investigation, M.D.P., A.N.D., A.A.O., M.V.K. and V.Y.Z.; writing—original draft preparation, M.D.P., M.V.M., N.Y.P., V.Y.Z. and I.V.O.; writing—review and editing, M.D.P., M.V.M.,

N.Y.P. and A.V.S.; project administration, M.D.P. and N.Y.P. All authors have read and agreed to the published version of the manuscript.

**Funding:** Tasks related to the study, development, and implementation of a research complex for investigating the properties of helical electron beams (including secondary emission effects), described in Sections 1 and 2, as well as the creation of microwave components using CMPS technology (throughout the paper), were carried out under the Russian Science Foundation grant 21-19-00884. Works described in Section 4 are supported by the Russian Science Foundation, Project No. 20-72-10116. Theoretical research and experimental test of Bragg resonators, described in Section 3, was supported by the Russian State Assignment Program, IAP RAS project 0030-2019-0027.

**Data Availability Statement:** Not applicable.

**Acknowledgments:** The authors of the work are grateful for the use of the CMPS technology, which is at the stage of patent registration.

**Conflicts of Interest:** The authors declare no conflict of interest.

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
