# Peer review of "Experimental Studies of Microwave Tubes with Components of Electron–Optical and Electrodynamic Systems Implemented Using Novel 3D Additive Technology"

_instruments, doi:10.3390/instruments6040081_

Round 1
Reviewer 1 Report
The matter treated in this paper and the proposed technology represents a significant breakthrough in the design and construction of gyrotron devices. The possibility of accurate control of the heat load opens new perspectives on the safe design of high-power microwave tubes.
Author Response
The authors are grateful to the reviewer for a careful reading of our article and a high appreciation of the results of our work.
Reviewer 2 Report
The article presents various experimental studies of microwave tubes with components manufactured with novel 3D additive technology.
In the introduction the state of the art of various gyro-devices, focusing on gyrotron components. It was noticed that all the relative references are either self-citations or originated from Russian research institutes. From my knowledge, significant contribution to gyrotrons has been made by European research groups, Japanese and Indian ones. Authors should provide sufficient citations to the aforementioned groups, otherwise the reader understands that the gyrotron research is only conducted by Russian research groups, which is not the case. As an example, significant research on multistage depressed collectors has been performed in IHM-KIT. Furthermore, authors should explain what the beam transport case is, in order to be clearer to the reader. Finally, in the introduction, a few references on the state of art of CMPS method are given and the authors are kindly advised to present it using relevant citations.
Section 2 presents the state of the art of additive technologies and specific features of the proposed CMPS technology. A lot of technical details are given which may confuse the reader and take him away from the subject of this work. The same is also holds for the next section. Authors are advised to rephrase the whole section and transfer the introduction of it to the article’s introductory section.
Section 3 presents the electron-optical system for technological gyrotron. The problem is described in detail, however there is a lack of references regarding the models accounting for secondary electron emission (other than that presented in the paper) and relative experiments, which would strengthen the authors’ arguments. In addition, authors should explain in detail the “structure of complex geometry, which suppresses the excitation of the RF-field in the beam transport channel”, in order to be more understandable for the reader the role of CMPS method and its novelty regarding the application. The rest of the section contains a lot of technicalities which I do not know if they add to the novelty of the application of CMPS method, and little to no details are given regarding the numerical simulations in the same time. Authors should be aware that lines 356-368 are duplicated in the next page, lines 373-385.
Section 4 refers to a Sub-GW Cherenkov oscillator with 2D-periodical SWS. The problem is presented in adequate detail, as well as the role of CMPS. However, authors should include axis labels in Figure 13. However, the whole section lacks a lot of CMPS relevant results and how this method contributes to the improvement of the existing structure. Without these results the section is just a presentation of irrelevant to the application of the CMPS method.
Section 5 presents a selective resonator for high-harmonic gyrotron. The basic theory is described in detail, however there is a significant lack of citations from line 550 tp 562. Why choosing just one corrugation and why the TEm,n+1 will be excited and not TEm,n-1? These and similar questions should be answered thoroughly in the manuscript. In Line 583 we read the following statement, if the TE1,1 is suppressed, the TE0,2 will be excited at the third harmonic. Authors should explain why this will happen in detail. Later in this section, technical details are given regarding the numerical simulations. The electron beam has spread in the pitch angle, which should be given. However, the major lack of this section is how CMPS is used here, and what are the advantages and disadvantages of its application. As with the previous section, without relevant results, the section is just a presentation of a selective resonator for high-harmonic gyrotron and out of the scope as expressed in the title of the manuscript.
As an overall impression, the manuscript lacks citations in key points and how the CMPS method is applied, its advantages and drawbacks. Furthermore, the sections seems to be somewhat “disconnected” one from each other. Authors are kindly advised to revisit the manuscript and make it clearer to the reader, where the application of CMPS method on key gyrotron component will be a lot more prominent.
Author Response
The article presents various experimental studies of microwave tubes with components manufactured with novel 3D additive technology.
In the introduction the state of the art of various gyro-devices, focusing on gyrotron components. It was noticed that all the relative references are either self-citations or originated from Russian research institutes. From my knowledge, significant contribution to gyrotrons has been made by European research groups, Japanese and Indian ones. Authors should provide sufficient citations to the aforementioned groups, otherwise the reader understands that the gyrotron research is only conducted by Russian research groups, which is not the case. As an example, significant research on multistage depressed collectors has been performed in IHM-KIT. Furthermore, authors should explain what the beam transport case is, in order to be clearer to the reader. Finally, in the introduction, a few references on the state of art of CMPS method are given and the authors are kindly advised to present it using relevant citations.
Our response:
We have edited the introduction by adding references to the results of European, Japanese and Indian colleagues, who undoubtedly made a very important contribution to gyrotron research.
We have also added a reference to IHM-KIT's work on a two-stage depressed collector that uses method of electron fraction separation by [E, B] crossed fields proposed by KIT scientists.
According to the CMPS technology, we have replaced on of the reference about it from the end next to the patent’s reference. Due to the technology is new many references is not still published, we can add references where it is applied, but it will lead to growth of self-citation.
Corresponding change in manuscript: Yes
Location of Change:
Section: I (Introduction), References
Page #1,2
Section 2 presents the state of the art of additive technologies and specific features of the proposed CMPS technology. A lot of technical details are given which may confuse the reader and take him away from the subject of this work. The same is also holds for the next section. Authors are advised to rephrase the whole section and transfer the introduction of it to the article’s introductory section.
Our response: We don’t think that such information may take reader away from subject of this work, because this is basement of creation elements described in this article. But due to reviewer’s request, we deleted this section and include the main information to the Introduction. Some information we replaced to sections, where this information is needed.
Corresponding change in manuscript: Yes
Location of Change:
Section: I, II
Page #2,3
Section 3 presents the electron-optical system for technological gyrotron. The problem is described in detail, however there is a lack of references regarding the models accounting for secondary electron emission (other than that presented in the paper) and relative experiments, which would strengthen the authors’ arguments.
Our response:
We have added references to other secondary emission models known to us, in particular, to the Furman model used in CST Studio. We have also mentioned that our model is simple and qualitative, we are currently study the influence of various model parameters on the energy distribution of spent electrons on the collector, which will subsequently be verified experimentally.
Corresponding change in manuscript: Yes
Location of Change:
Section: II, References
Page #3, bottom
In addition, authors should explain in detail the “structure of complex geometry, which suppresses the excitation of the RF-field in the beam transport channel”, in order to be more understandable for the reader the role of CMPS method and its novelty regarding the application.
Our response: Novelty of CMPS technology is not connected to novelty of geometry and mechanism of suppression the excitation of the RF-field in the beam transport channel. The profile of this element is the know-how of the IAP RAS, and we would not like to talk about it in detail, mentioning only the main property of this system – the low probability of excitation of electromagnetic oscillations in the beam transport channel. This element doesn’t play an important role in the presented research. A well as the other elements, this one can be made by common CNC machining technology, but it will be much more expensive and time taking. To suppress the excitation, this element can be done only by polymer (without further metallization), but it is not suitable for high vacuum condition with presence of 55-kW electron beam.
Corresponding change in manuscript: Yes
Location of Change:
Section: II
Page #6,7
The rest of the section contains a lot of technicalities which I do not know if they add to the novelty of the application of CMPS method, and little to no details are given regarding the numerical simulations in the same time.
Our response: In this chapter real examples of technology of manufacturing all-copper elements with complicated form with internal channels are presented. Elements made using this technology are by an order of magnitude less expensive and faster to manufacture. Their performance has been shown by the example of part of real collector of the gyrotron with integrated thermal sensors. An alternative to the additive manufacturing of all-metal hollow elements are technologies such as SLS and SLM. Such machines cost millions of dollars, have great difficulty in working with copper, and also have significantly higher surface roughness, which can lead to manufacturing errors. Also, in this chapter a lot of simulation results (thermal distribution in different types and elements of collector under electron beam deposition) are shown, that helped us to choose the best way of optimization a collector manufacturing process.
Authors should be aware that lines 356-368 are duplicated in the next page, lines 373-385.
Our response: Authors apologize and thank the reviewer for flagging this mistake. It happened during filling the journal template.
Corresponding change in manuscript: Yes
Location of Change:
Section: II
Page #8
Section 4 refers to a Sub-GW Cherenkov oscillator with 2D-periodical SWS. The problem is presented in adequate detail, as well as the role of CMPS. However, authors should include axis labels in Figure 13. However, the whole section lacks a lot of CMPS relevant results and how this method contributes to the improvement of the existing structure. Without these results the section is just a presentation of irrelevant to the application of the CMPS method.
Our response: We have made appropriate corrections to Fig. 13. This section is devoted to demonstration of the operability of the CMPS technology for small-scale selective elements of high-power short-wavelength Cherenkov oscillator. The proof-of-principle experiments with Ka-band surface-wave oscillators based on periodical structures fabricated using this technology have been carried out. The conducted experimental studies and preliminary estimates show the possibility of using the proposed technology for vacuum tubes in the sub-terahertz range.
Corresponding change in manuscript: Yes
Location of Change:
Section: III
Page #12, right column, Fig. 13.
Section 5 presents a selective resonator for high-harmonic gyrotron. The basic theory is described in detail, however there is a significant lack of citations from line 550 tp 562. Why choosing just one corrugation and why the TEm,n+1 will be excited and not TEm,n-1? These and similar questions should be answered thoroughly in the manuscript. In Line 583 we read the following statement, if the TE1,1 is suppressed, the TE0,2 will be excited at the third harmonic. Authors should explain why this will happen in detail. Later in this section, technical details are given regarding the numerical simulations. The electron beam has spread in the pitch angle, which should be given. However, the major lack of this section is how CMPS is used here, and what are the advantages and disadvantages of its application. As with the previous section, without relevant results, the section is just a presentation of a selective resonator for high-harmonic gyrotron and out of the scope as expressed in the title of the manuscript.
Our response:
This is a novel method for improving the mode selection in gyrotron. This is described in detail in paper [21], and we can not add here any more citations.
Why choosing just one corrugation and why the TEm,n+1 will be excited and not TEm,n-1?
If the radius of the main part of the cavity corresponds to a near-cutoff mode TE m,n at a given (operating) frequency, then the radius of the inhomogeneity corresponds to the mode TE m, n+1 at the same frequency. In order to clarify this fact, the first paragraph in Sect. V is changed as follows (the changed parts of the text are in red):
It is important that such inhomogeneities have a resonant character only for one operating mode. The meaning of the term "resonant character" used here is illustrated in Fig. 15 a. The azimuthal component of the resonator's own mode is zero on its wall. This means that if the operating mode of the gyrotron is the near-cutoff mode ТЕm,n, then the radius of the inhomogeneity is chosen so that it corresponds to the near-cutoff mode ТЕm,n+1 at the same frequency. In fact, an increase in the radius in the inhomogeneity corresponds to one additional radial variation of the wave field at a given frequency. As a result, the operating ТЕm,n mode is transformed inside the inhomogeneity it into the ТЕm,n+1 without losses and reflections. Therefore, the operating mode practically “does not feel” this inhomogeneity, and the excitation of this wave occurs in the same way as in a regular circular cavity. In fact, the cavity with “proper” azimuthal inhomogeneities respesent a resonant structure for a single pair of modes ТЕm,n + ТЕm,n+1.
At the same time, for parasitic waves, the presence of this inhomogeneity leads to the fact that the cavity ceases to have the property of azimuthal symmetry (Fig. 15 b). As a result, a number of factors (stopping wave rotation and converting a circularly polarized wave into two standing waves, distortion of the wave transverse structure, and an increase in Ohmic losses) leads to a noticeable increase in the starting currents of the parasitic waves [21].
As for just one corrugation, the following text is added to Sect. V:
Note that in the scheme described in [21] a cavity with two azimuthal irregularities was proposed. The fact is that, as a rule, an azimuthally asymmetric transverse mode with the "correct" direction of its rotation is used as an operating mode in gyrotrons. In this case, the two grooves (located in places corresponding to the zero and maximum of the field of the standing component of the rotating operating mode) are needed in order to prevent the rotation of the operating wave from stopping. In contrast to work [21], in this paper we use only one groove to improve the mode selection. This is a consequence of the advantage of the operating azimuthally-symmetric mode TE0,2 we have chosen, that there is no problem of stopping rotation for it.
In Line 583 we read the following statement, if the TE1,1 is suppressed, the TE0,2 will be excited at the third harmonic. Authors should explain why this will happen in detail.
The corresponding text in Sect. V is changed as follows (the changed parts of the text are in red):
However, if the excitation of the mode TE1,1 is suppressed, then we can expect the excitation of other modes, in particular, the mode TE0,2 at the third cyclotron harmonic. Actually, according to Fig. 16, in the magnetic field range close to 6.9 T in this case only the mode TE0,2 can be excited at operating currents 1-2A. In the absence of the mode TE1,1, she has no other competitors. The numerical simulations below confirm this statement.
The electron beam has spread in the pitch angle, which should be given.
The corresponding text in Sect. V is changed as follows (the changed parts of the text are in red):
The electron energy in the input electron beam was 23 kV, the pitch factor was g = 1.5 with spread in electron velocities taken into account (the width of the Gaussian transverse velocity distribution function is 20%, which is typical for this type of gyrotron guns).
Corresponding change in manuscript: Yes
Location of Change:
Section: II
Page #2, right column, Fig. 2.
However, the major lack of this section is how CMPS is used here, and what are the advantages and disadvantages of its application.
The gyrotron cavity is the most sensitive part of the generator in terms of manufacturing error. In the standard form, the gyrotron cavity has a simple cylindrical shape, which greatly simplifies the process of its manufacture on a CNC machine. If it is necessary to create a resonator with an azimuthally asymmetric shape, the technology changes significantly. It is necessary to make a negative of a real shape from aluminum with high accuracy, then build up a thick layer of copper, perform surface treatment and etch the metal blank. All this is significantly inferior to the process of creating such an element using the CMPS method.
As an overall impression, the manuscript lacks citations in key points and how the CMPS method is applied, its advantages and drawbacks. Furthermore, the sections seem to be somewhat “disconnected” one from each other. Authors are kindly advised to revisit the manuscript and make it clearer to the reader, where the application of CMPS method on key gyrotron component will be a lot more prominent.
Our response: In this paper we showed the first experiments that proof the possibility of use CMPS technology in the field of high-power microwave sources, both the cases of radiation and beam power. Thus, in this article we investigated the high-power electrodynamic element of Cherenkov oscillator and part of electron-optical system of the gyrotron (also touched the problem with irregular cavity).

Round 2
Reviewer 2 Report
The authors addressed all aspects of the initial review in detail and clarity. They answered all the questions in detail and added appropriate text parts in the manuscript. The changes improved significantly the quality of the manuscript. Now, the latter is clearer to the reader and more robust.